# The Limitations of a Hypothetical All-Variant COVID-19 Vaccine: A Simulation Study

**DOI:** 10.3390/vaccines12050532

**Published:** 2024-05-13

**Authors:** Robert J. Kosinski

**Affiliations:** Independent Researcher, 303 Tamassee Drive, Clemson, SC 29631, USA; rjksn@clemson.edu

**Keywords:** SARS-CoV-2, COVID-19, compartmental model, epidemiology

## Abstract

This paper simulates a hypothetical pan-coronavirus vaccine that confers immediate sterilizing immunity against all SARS-CoV-2 variants. Simulations used a SEIIS (susceptible, exposed, infective, immune, susceptible) spreadsheet model that ran two parallel subpopulations: one that accepted vaccination and another that refused it. The two subpopulations could transmit infections to one another. Using data from the United States (US), the simulated vaccine was tested against limiting factors such as vaccine hesitancy, slow vaccination distribution, and the development of high-transmission variants. The vaccine was often successful at reducing cases, but high-transmission variants and discontinuation of non-pharmaceutical interventions (NPIs) such as masking greatly elevated cases. A puzzling outcome was that if NPIs were discontinued and high-transmission variants became common, the model predicted consistently higher rates of disease than are actually observed in the US in 2024. However, if cumulative exposure to virus antigens increased the duration of immunity or decreased the infectivity of the virus, the model predictions were brought back into a more realistic range. The major finding was that even when a COVID-19 vaccine always produces sterilizing immunity against every SARS-CoV-2 variant, its ability to control the epidemic can be compromised by multiple common conditions.

## 1. Introduction

The production of vaccines against the SARS-CoV-2 virus was a landmark in a pandemic that has caused over 6.9 million hospitalizations and 1.18 million deaths in the US as of 13 March 2024 [1]. Early in the pandemic, vaccines were a powerful public health tool against COVID-19. Moutinho [2] described how the town of Serrana, Brazil, vaccinated 27,000 of the 45,000 residents with CoronaVac and achieved an 80% drop in symptomatic COVID-19 cases. Balicer and Ohana [3] reported that Israel achieved a 100-fold drop in new cases through an energetic campaign that vaccinated 2.5% of the population on some days. 

The vaccination effort in the US has been extensive. As of 11 May 2023, 69.5% of the population had received the primary series of the vaccine (including 94.4% of those over 65), and at least one bivalent booster dose had been given to 17% of the population [4]. By the third quarter of 2022, 96.4% of Americans over the age of 16 had SARS-CoV-2 antibodies and 73.8% had vaccine-induced antibodies [5]. The number of completed vaccinations/day peaked at about 4.6 million on 1 April 2021 (about 1.4% of the whole US population on that one day). However, by 3 May 2023, vaccinations had fallen to 50,000 vaccinations/day [4], less than 1% of the peak vaccination rate. 

Although the quick development of high-efficacy vaccines was a scientific triumph [6], producing the vaccine is only part of the public health battle. Aschwanden [7] summarized reasons why herd immunity would be difficult to achieve through vaccination: vaccines might still allow disease transmission, vaccine rollouts in different countries have been slow and uneven, there was (at the time) still no vaccine for children, pockets of vaccine hesitancy would serve as reservoirs of disease, new variants will develop, vaccine-induced immunity would expire, and the arrival of vaccines might cause the public to discontinue non-pharmaceutical interventions (NPIs) such as masking. 

### 1.1. Previous COVID Vaccination Simulation Studies

Simulation has a long history in epidemiology. Kermack and McKendrick [8] laid the foundation in 1927, proposing a system of differential equations that would allow the prediction of how an epidemic of a communicable disease would spread, and then possibly stop its spread with some individuals still uninfected. This became the basis for the calculation of the degree of vaccination required to establish herd immunity.

More recently, COVID-19 has had an active simulation history. Several studies from the period before the completion of the first COVID-19 vaccine used simulations to anticipate the effect the new vaccine would have on the pandemic. Saad-Roy et al. [9] investigated some of the same questions considered by this study. Using a model that allowed reinfection after immunity had expired, they simulated the five-year course of COVID-19 in a northern, temperate location such as New York, with seasonal peaks in the fall, partial suppression of spread by non-pharmaceutical interventions (NPIs), and the inception of vaccination 1.5 years after the beginning of the epidemic. They found that high vaccine efficacy, fast rates of vaccine administration, and long durations of vaccinal immunity had the greatest chance of ending the epidemic. They found that vaccine “refusers” could reduce overall immunity and allow outbreaks despite vaccination.

Patel et al. [10] described an agent-based model of a vaccination campaign in North Carolina, including different races and ethnicities, and simulated 75% vaccine coverage of the population in six months. They found that high vaccine coverage with a low-efficacy vaccine was better than low coverage with a high-efficacy vaccine and that abandoning NPIs while the vaccines are distributed “may result in substantial increases in infections, hospitalizations, and deaths”.

Moghadas et al. [11] described an agent-based model of the US that predicted a vaccine with a maximum 70% coverage in any age group (delivered at about 2% of the population per week) could substantially reduce hospitalizations and death, even if the vaccine had only limited ability to prevent infection. They cautioned that maintenance of NPIs would be necessary to achieve these effects. 

Gozzi et al. [12] simulated different vaccination strategies in six diverse countries ranging from Egypt to Canada. Vaccinal immunity did not expire during the length of their simulations. Their focus was on the role of NPIs as vaccinations were rolled out. They concluded that the “great effort” of vaccination would be wasted if NPIs were abandoned too soon. As a matter of fact, a slow vaccination speed could have the paradoxical effect of increasing deaths because it was ineffectual in conferring immunity, but still caused the public to abandon NPIs. 

A long-running simulation effort of a different kind was described by Howerton et al. [13]. The US COVID-19 Scenario Modeling Hub made detailed but short-term predictions of outbreaks. The authors concluded that well-calibrated models could make accurate projections for periods shorter than 22 weeks, but after that, the unanticipated arrival of new variants degraded accuracy. 

### 1.2. This Report

Altmann and Boyton [14] discussed strategy options against COVID-19, and one option would be to focus on the development of a “pan-coronavirus” vaccine that would be effective against all variants. This proposal suggests the subject of this paper: If such a vaccine existed, what is the best COVID-19 control that could be expected from it? The answer depends on at least six variables beyond the efficacy of the vaccine: the degree of vaccine hesitancy, the speed of vaccine uptake by the population, the persistence of desire to get boosters, the duration of vaccine-induced immunity, the persistence of NPIs such as masking, and the evolution of high-transmission variants. Although the hypothetical vaccine itself may be instantly effective against all variants, it is possible that ancillary factors like the rapid waning of vaccinal immunity might compromise the vaccine’s success against the pandemic.

In this paper’s simulation scenario, a COVID-19 epidemic starts when an infected individual enters a population at 0 years, and the epidemic proceeds with control only by NPIs for a year. At that point, a “universal” COVID-19 vaccine is developed and distributed. This vaccine:(a)produces immediate, sterilizing immunity against all SARS-CoV-2 variants;(b)induces vaccinal immunity that lasts for a defined period. During this time, no vaccinated individual can be reinfected. However, immunity wanes exponentially with time.

The long-term success of the vaccine is judged by the “post-vaccine case rate”, or the number of COVID-19 cases per thousand per year over the period from one year to five years after the introduction of the vaccine (the timespan covered by the red rectangle in Figure 1). The more the case rate was reduced over these four post-vaccine years, the more effective the vaccination regime was judged to be. In rough terms, the model simulates the US experience of 2020 (initial outbreak controlled by NPIs only), introduction of the vaccine and the initial vaccination campaign (2021), and then the four-year “post-vaccine era” of 2022–2025. It draws its conclusions from case rates in the “2022–2025” period. 

At the height of the US COVID-19 pandemic, Dr. Anthony Fauci suggested in popular press interviews that the US should aspire to fewer than 10,000 infections/day [15]. This equates to approximately 10 cases per thousand per year in the US, giving an average individual a 1% chance of contracting COVID-19 each year. Therefore, an average post-vaccine case rate of 10 cases/thousand/year or lower will be regarded as a success. 

The variables tested for their effect on the post-vaccine case rate were:(a)the percent of the population willing to accept vaccination (25–100%);(b)the speed of vaccine distribution, measured by the percent of susceptibles vaccinated per day (0–2.5%);(c)maintenance or cessation of NPIs such as masking;(d)a weakening of interest in getting booster vaccinations;(e)the duration of vaccine-induced immunity (50–750 days);(f)the development of a new variant with a higher R_0_ (changing from a “2020” base value of 2.87 to “2021–2024” values of 3.5, 5.0, or 10.0).(g)a decrease in the ability of the virus to start an infection due to long-term exposure to viral antigens;(h)a lengthening of the duration of immunity due to long-term exposure to viral antigens.

## 2. Materials and Methods

The simulations used an SEIIS compartmental model using difference equations on a Microsoft Excel spreadsheet. The spreadsheet (Kosinski COVID Simulation Spreadsheet.xlsx) is available in Appendix A. The following paragraphs describe the model without vaccination or NPIs first, then the model’s simulation of NPIs, and finally the simulation of vaccination.

In a compartmental model, the population is divided into a series of groups (the compartments) with equations governing transfer between groups. Each group is internally homogeneous (e.g., all susceptible individuals are the same and all immune individuals are the same). The SEIIS model contained four different groups of individuals in a model population of one million: Susceptible (able to catch the disease), Exposed (infected but not yet able to infect others), Infective (able to infect others), and Immunes. Because the model did not include mortality and because a patient who has recovered from COVID-19 can contract it again after the period of immunity is over, the final “S” is a return to the susceptible compartment. The overall model structure was a cycling of individuals between susceptibles and immunes. The durations of these stages were as follows:Susceptible → Exposed (5 days) → Infective (5 Days) → Immune (variable time) → Susceptible

The duration of immunity was the same for both disease-induced immunes and vaccination-induced immunes. The duration of immunity mainly determined how fast the individuals cycled between the compartments above.

Each exposed, infective, and immune individual moved one day at a time through its compartment. For example, an exposed individual was moved from day one of exposure to day two, then to day three, etc. Although “Exposed” and “Infective” groups were handled separately in the calculations, they are combined into one group called “Infected” in Figure 1.

If we consider only one population and ignore corrections for the use of NPIs (Equations (9) and (10)), the number of new cases per day is given by
new cases/day = (R_0_/5)*inf*[sus/(sus + exp + inf + imm)](1)
where inf = the number of infectives, exp = the number of individuals who are exposed to the disease but are not yet infective, sus = the number of susceptibles, and imm = immunes. The “5” refers to the five-day infective period. R_0_ is the basic reproduction number, the number of cases each case can start during its infective period in a population consisting completely of susceptibles. R_0_ represents the maximum transmissibility of the disease. The initial R_0_ was assumed to be 2.87 early in the epidemic [16] and throughout the first set of simulations. In the second set of simulations that dealt with high-transmission variants, R_0_ from 600 days or later in the simulation could take values of 3.5, 5.0, or 10.0 [17,18]. 

Let the proportion of susceptibles in a population (the quantity in brackets in Equation (1)) be psus. If we consider only one population and ignore corrections for the use of NPIs (Equations (9) and (10)), the number of new cases per infective during the infective period is given by
new cases per infective over infective period = R_0_*psus(2)

Therefore, when the fraction of susceptibles in a population is less than 1/R_0_, the new cases per infective are less than 1.0 during the whole infective period, and each infective will not replace itself during its five-day lifespan. Consequently, new cases per infective must decline. Getting R_0_*psus below 1.0 is the basis of herd immunity.

The model had the capability of running two populations on two different sheets. Population 1 accepted vaccination and Population 2 refused it. These two populations mixed randomly with one another. These populations were then combined on a third sheet. This allowed simulation of two populations that had different characteristics, but which interacted with each other by “sharing” infective and susceptible individuals.

Let the total number of individuals in Populations 1 and 2 be pop_1_ and pop_2_. Where the number of infectives in Population 1 in an iteration is inf_1_ and the number of infectives in Population 2 is inf_2_, then the total number of infectives “seen” by Populations 1 and 2 is the same:inf_tot_ = inf_1_ + inf_2_(3)

The importance of the proportion of susceptibles (psus) in a population is seen in Equation (2). The proportion of susceptibles in Population 1 and Population 2 will be
psus_1_ = sus_1_/pop_1_(4)
psus_2_ = sus_2_/pop_2_(5)

Two other variables, not to be confused with psus_1_ and psus_2_, are susfrac_1_ and susfrac_2_. Susfrac_1_ is the fraction of all the susceptibles (in both populations) that are in Population 1. Susfrac_2_ is the corresponding fraction for Population 2. For example, say that at the beginning of the simulation, both Population 1 and Population 2 are all susceptibles, but ¾ of all the individuals are in Population 1. Then both psus_1_ = psus_2_ = 1.0, but susfrac_1_ = 0.75 and susfrac_2_ = 0.25. That is,
susfrac_1_ = sus_1_/(sus_1_ + sus_2_)(6)
susfrac_2_ = sus_2_/(sus_1_ + sus_2_)(7)

The susfrac variables are used to allocate the new cases to Population 1 and Population 2. The number of new infections in a day in Populations 1 and 2 will be
newinf_1_ = (R_0_/5)*inf_tot_*psus_1_*susfrac_1_(8)
newinf_2_ = (R_0_/5)*inf_tot_*psus_2_*susfrac_2_(9)
where the “5” refers to the five-day infective period. The “psus” factor reduces the number of new cases as the proportion of susceptibles in populations 1 and 2 falls, and will eventually reach zero if there are no susceptibles left in a population. But early in the epidemic, psus_1_ and psus_2_ will be close to 1.0, and every infective in populations 1 and 2 will produce R_0_/5 new cases per day. The susfrac variable allocates the new infections to Population 1 and Population 2. If a population has no susceptibles left, no new infections would be allocated to it. For example, if Eq. 3 determines that on a certain day there are 100 new cases of disease and Population 1 has 10% of all the susceptibles, Population 1 will get 10 new cases and Population 2 will get 90 new cases.

New infections delete members from the susceptible compartment and add them to day 1 of the exposed compartment. They then proceed through the exposed, infective, and immune compartments, day by day, until they return to the susceptibles after their period of immunity has expired. Vaccination takes a fraction of the susceptibles each day, removes them from the susceptible compartment, and adds them directly into the immune compartment. The duration of immunity is explained in more detail below.

There were no deaths, births, or demographic groups in the model (aside from Population 1 and Population 2). There was no seasonal change in infectivity. There was also only one variant, but in some simulations, the R_0_ was changed to simulate the arrival of a new, more contagious variant.

A single-population simulation started when one individual at the beginning of their five-day infective career entered the population of 999,999 susceptible individuals. If two interacting populations were being used, each population got one infective on the first day. The sizes of Populations 1 and 2 were adjusted so their sum was 1,000,000 (e.g., if Population 1 was 750,000, Population 2 was 250,000). 

### 2.1. Non-Pharmaceutical Interventions

In the first year of each simulation, there was no vaccination, and all control of the epidemic relied on strategies like social distancing, masking, and hand-washing (non-pharmaceutical interventions, or NPIs). NPIs were simulated by multiplying R_0_ by (1 − lkd). Lkd was a “lockdown” parameter that reached a maximum of 0.5, meaning social interaction and disease transmission were reduced by 50%. R_0_ was nearly undiminished when the infected percent of the population was low and reached 0.5*R_0_ when the infection rate was high. Lkd followed Michaelis–Menten kinetics (Equations (10) and (11)). For population 1,
infected%_1_ = 100*(exposed_1_ + inf_1_)/pop_1_(10)
lkd_1_ = 0.5*infected%_1_/(infected%_1_ + 0.01)(11)

There were similar equations for population 2. Populations 1 and 2 had the same lockdown equations and the same maximum lockdown parameter of 0.5. Equation (11) allowed the disease to spread rapidly at first. However, once the infected percentage of the population increased beyond 0.01%, lkd_1_ and lkd_2_ rapidly rose to 0.5.

### 2.2. Waning of Immunity

Kosinski [19] used a similar model to investigate the effect of the duration of immunity on a COVID epidemic. The results were that short durations of immunity produced repeated peaks and increased the average number of cases per person. That paper used what might be called an “age-structured” model of immunity. If the duration of immunity is assumed to be one year, then every individual who enters immunity on a certain day exits from immunity exactly one year later. This tends to produce repeated fluctuations in the prevalence of disease as immunity in the population expires at the same time.

The more traditional SEIIS compartmental model (used here) treats all the immune individuals (no matter when they became immune) as one pool from which a certain fraction (∂) leaves each iteration to re-join the susceptibles. If there were no new inputs into the immune compartment, this means that the immune population would experience an exponential decline and that the average duration of immunity would be 1/∂. For example, for an average duration of immunity of one year, 1/365 of the immunes would leave the immune compartment each day. Therefore, 1/365 = ∂, and the average duration of immunity would be 1/∂ = 365 days. The key to estimating the average duration of immunity in this model is to determine the fraction of immunity that is lost per day (∂). As will be seen, the average duration of immunity has a strong impact on the ability of the vaccine to control the disease.

The mean duration of COVID-19 immunity conferred by COVID-19 vaccines seems to be much shorter than a year. Link-Gelles et al. [20] determined that effectiveness against hospitalization of the bivalent mRNA vaccine in adults declines from 62% at a mean of 33 days to 24% at a mean of 150 days (117 days later). Assuming exponential decline, a decline from 62% effective to 24% effective is a decline to 38.7% of the initial effectiveness value (24/62 = 0.387). Ln(0.387) = −0.949, and because the decline period was 117 days, the ln of the persistence/day = −0.949/117 = −0.00811. This implies that the immune persistence/day = e^−0.00811^ = 0.9919 and therefore the loss/day is 1 − 0.9919 = 0.0081. The reciprocal of 0.0081/day is approximately 124 days = 1/∂ = the mean duration of effectiveness. Therefore, the exponential model using these data predicts that immunity against hospitalization is losing about 1/124 of its effectiveness/day.

For comparison, Menegale et al. [21] conducted an extensive literature review and reported in their Figure 1 that immune effectiveness against symptomatic disease for a first vaccination against the omicron variant falls from about 55% at one month post-vaccination to 18% at six months (a period of 150 days). In other words, it has lost 67% of its initial effectiveness in 150 days, and its mean duration of effectiveness (1/∂) is 135 days. The same figure shows that immunity against symptomatic omicron disease conferred by a booster vaccination declined from 45% to 23% effective in 150 days, a loss of 49% of its initial effectiveness, corresponding to a mean duration of booster effectiveness of 224 days. 

DeCuir et al. [22] examined the duration of immunity induced by the monovalent XBB.1.5 vaccine against emergency department and urgent care visits. Their data allows computation of a mean duration of immunity of 291 days for adults 18–64 years of age and 146 days for adults over 65.

This paper will use 135 days as the default average duration of immunity induced by both disease and vaccination. This duration will be a constant in most simulations. 

### 2.3. Vaccination

Population 2 was never vaccinated. In Population 1, a percentage (ranging from 0 to 2.5% per day) of the susceptibles were moved into the immune population every day. Vaccination rates used in most simulations were 0%, 0.25%, 0.5%, 0.74%, and 1.0%/day. 2.5%/day was used in a few cases. Vaccination produced an exponential decline in the number of individuals in the susceptible compartment. The simulated vaccine always created same-day sterilizing immunity against all variants. 

### 2.4. Maintenance of NPIs

Because the advent of the vaccine could reduce the willingness of the population to maintain NPI use, 500 days from the start of the simulation (about “15 May” of “2021”), the “lockdown” parameter either remained at 0.5, was reduced to 0.25, or was reduced to zero (complete cessation of NPIs). This was meant to simulate the CDC announcement on 13 May 2021 that vaccinated individuals no longer needed to wear masks indoors or social-distance from others, which might have encouraged the unvaccinated to do likewise.

### 2.5. High-Transmission Variants

All of the simulations above were conducted first with a pathogen with a constant R_0_ of 2.87 [16] because the inclusion of more transmissible variants so elevated case rates that it tended to hide every other effect. After the “2.87” simulations (Table 1, Table 2, Table 3, Table 4 and Table 5), a reduced set of simulations (Table 6, Table 7, Table 8, Table 9, Table 10, Table 11 and Table 12) was conducted with pathogens that started with an R_0_ of 2.87 but then increased their R_0_ to 3.5, 5.0, or 10.0 between 500 and 600 days (“15 May” to “23 August” of the year the vaccine was introduced). The R_0_ then stayed at this new value for the rest of the simulation.

### 2.6. Case Rates

The main response variable was the number of COVID-19 cases per thousand individuals per year (starting one year after vaccination began). The highest one-day case rate ever seen in the US epidemic so far occurred at the peak of the Omicron surge on 11 January 2022, when more than 1.3 million new cases were reported [1]. If this daily rate of new cases continued for a year, it would be 1438 cases/thousand/year. However, by late April of 2023, the average reported US case rate had fallen to 119,000 cases per week (only 19 cases/thousand/year) [1]. 

The CDC stopped reporting case rates in May of 2023. However, the CDC Website is still reporting hospital admissions as of April 2024. Using screen captures of the CDC Data Tracker site taken during the period from 30 April 2022 to 30 April 2023 (the most recent year in which both case rates and hospital admissions were presented), it was possible to estimate that during that 2022–2023 period, approximately one case in 14.46 resulted in hospital admission. This ratio allowed a rough estimate of case rates from the hospital admission data up to the end of March 2024.

### 2.7. Reduction in Infections Due to Exposure to Virus Antigens

The results will disclose that the SEIIS model as described above predicts 2024 case rates that are much higher than those probably being experienced in the US. In order to explain the discrepancy, this paper will hypothesize that long-term exposure to virus antigens either strengthens the human immune response to reduce the chance of infection or increases the duration of immunity.

For the reduction in the chance of infection, the model recorded the cumulative number of infections and vaccinations and multiplied R_0_ by a factor that declined as the cumulative exposure to virus antigens increased (Equation (12)). The exposure to virus antigens was recorded as the percent of the population that had been either infected or vaccinated since the beginning of the simulation. Multiple infections or vaccinations could cause this percentage to go over 100%. The effect of this percentage was translated into a fraction using a parameter called Exposure_50_:R_0_ Reduction = Exposure_50_/(Exposure_50_ + % Exposure)(12)

Note that if % Exposure is zero, R_0_ Reduction = 1, and infection is undiminished. If % Exposure = Exposure_50_, R_0_ Reduction = 0.5. If % Exposure is much greater than Exposure_50_, R_0_ Reduction approaches zero, meaning that the ability of the virus to infect is reduced. The smaller Exposure_50_ is, the more previous exposure reduces the ability of the virus to infect.

When immune duration was increased by exposure to virus antigens, for every percent of either Population 1 or Population 2 that contracted a case or was vaccinated, either 0.01 or 0.05 days was added to the mean duration of immunity for that population. This caused immunity to wane more and more slowly as the simulation went on.

Neither R_0_ reduction nor lengthening of immune duration were used in Table 1, Table 2, Table 3, Table 4, Table 5, Table 6 and Table 7. 

## 3. Results

### 3.1. Sizes of Population 1 and Population 2 and the Speed of Vaccination in the US

The CDC’s cumulative data on fully vaccinated individuals [4] allows a retrospective estimate of the true sizes of Population 1, Population 2, and the vaccination rate during the US pandemic. First, by August of 2022, the cumulative number of completed vaccinations was nearly static at 223 million in a population of 330 million. Assuming that only Population 2 remains unvaccinated at this point, this estimates Population 1 as 68% of the US total, and Population 2 as 32%. During the period of rapid vaccination from 1 January 2021 to 30 June 2021 (181 days), the rate of decline in unvaccinated people was consistent with a constant vaccination rate of 0.74% of Population 1 per day (about 0.5% of the whole population per day). Vaccination rates in the rest of the article will be presented in terms of Population 1 because only Population 1 was vaccinated. The values above (Population 1 = 68%, Population 2 = 32%, vaccination rate = 0.74%/day) were used as default values, but a range of vaccination rates and Population 1 sizes were used in the simulations. 

### 3.2. Review of the Simulation Scenario

Recall that this paper’s scenario is that a COVID-19 epidemic starts at year zero in Figure 1, and during the first year, it is mitigated only by NPIs such as masking and social distancing. Then, at the one-year anniversary of the start of the epidemic, a vaccine is introduced and distributed for the remainder of the simulation. In some simulations, the interest in getting vaccinated was allowed to decline over time. The main responding variable is the number of new infections per thousand per year in years 2, 3, 4, and 5, indicated by the red rectangle in Figure 1. 

### 3.3. General Simulation Outcomes

Figure 1 contrasts the typical course of an unmitigated epidemic, the effect of NPIs on this outcome, and then the additional effect of vaccination. The purpose of this figure is to show the general form of model predictions, not to try to fit data from the US epidemic.

**Figure 1 vaccines-12-00532-f001:**
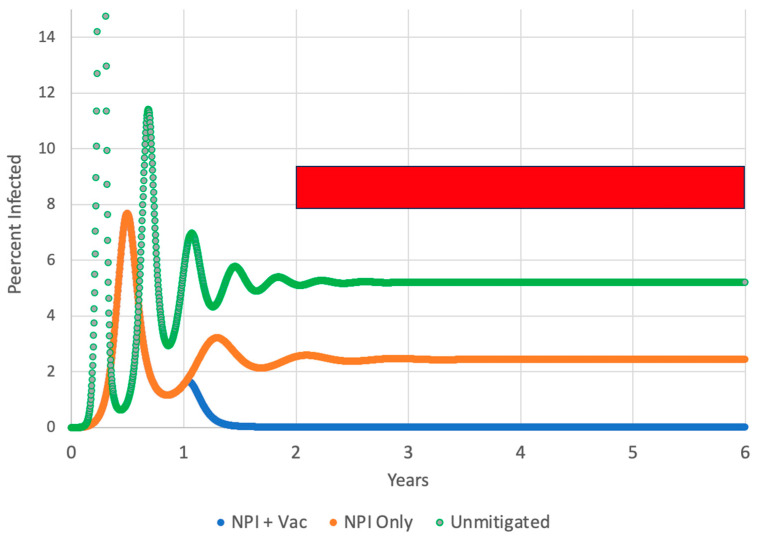
The course of the epidemic for R_0_ = 2.87 and a constant average duration of immunity of 135 days. The curves show an unmitigated epidemic (green), an epidemic controlled only by NPIs such as masking (orange), and an epidemic controlled by NPIs and by vaccination that starts at one year (blue). In this vaccination simulation, all members of the population were willing to accept the vaccine. The red rectangle indicates the “post-vaccine era” for which case rates were computed.

In the completely unmitigated epidemic (green curve), infection rose to a maximum of 40% of the population at the first peak, but any points above 15% are not shown in Figure 1 in order to avoid compressing the “NPI Only” and “Vac + NPIs” curves. The blue curve and the orange curve employed NPIs that could produce a 50% reduction in R_0_. These NPIs were active from the beginning of the simulation. In addition to the NPIs, the blue curve also used vaccination (delivered at 0.74% of the susceptible population per day) starting one year after the first case. For Figure 1 only, all members of the population accepted the vaccine (there was no vaccine-refusing Population 2). 

If there were no changes to parameters during a simulation, the curve showing the percent infection underwent an initial oscillation caused by a disease outbreak and building immunity but then settled down to a constant rate of new cases that were in equilibrium with the rate of expiration of immunity. More contagious disease or faster expiration of immunity led to a higher equilibrium rate of infection in the right portion of the curves. The equilibrium percentages of infection were approximately 5.2% of the population for the unmitigated epidemic, 2.1% for the NPIs-only scenario, and 0.015% for the NPIs plus vaccination scenario.

### 3.4. Vaccination Speed and Percent of Vaccine-Compliant Individuals

Given that the efficacy of the simulated vaccine is always 100%, the most important vaccination variables are the speed of vaccination (percent of Population 1 susceptibles vaccinated per day), and the percent of Population 1 (the population that accepts vaccination). 

**Table 1 vaccines-12-00532-t001:** The effect of vaccination speed and percentages of individuals willing to accept the vaccine on case rates. Variable shown is cases/thousand/year in years 2–5 in the combined population (1 + 2). R_0_ = 2.87, average duration of immunity = a constant 135 days, lockdown parameter 0.5 for Populations 1 and 2.

Vac/Day	Percent of Population 1 (Vaccine-Compliant)				
	100%	75%	68%	50%	25%
0%	773.6				
0.10%	540.0	602.1	619.0	661.6	718.7
0.25%	209.5	369.0	411.5	515.6	649.6
0.50%	17.2	89.0	145.9	326.5	562.1
0.74%	5.6	24.3	42.5	198.1	501.2
1.00%	2.1	11.5	19.0	110.2	452.2

### 3.5. Average Duration of Immunity

For any vaccination rate, the success of the vaccine was markedly enhanced by a long duration of immunity:

**Table 2 vaccines-12-00532-t002:** Post-vaccine case rates for immunity with constant average durations from 50 to 1000 days. The population is 68% Population 1 (vaccine-compliant) and 32% Population 2. R_0_ = 2.87, lockdown parameter = 0.5.

Vac/Day	Average Duration of Immunity (Days)				
	50	135	250	500	1000
0%	1851.1	773.6	424.4	236.0	111.6
0.10%	1709.7	619.0	281.4	88.5	16.0
0.25%	1504.1	411.5	111.8	16.2	3.2
0.50%	1179.9	146.0	20.6	4.4	0.5
0.74%	893.0	42.5	9.2	2.0	<0.1
1.00%	611.8	19.0	5.2	1.0	<0.1

### 3.6. Abandoning NPIs

Recall that in all the simulations above, the vaccines were combined with NPIs that produced a maximum lockdown parameter of 0.5 in both Population 1 and Population 2. This reduced R_0_ by a factor of 0.5 and resulted in much lower rates of infection when the R_0_ was 2.87, as in all simulations so far. However, as vaccination becomes more common and rates of disease drop, there might be a tendency for the public to stop masking and social distancing because they think that vaccination has replaced NPIs. In Table 3, Populations 1 and 2 keep their maximum lockdown parameter at 0.5 until day 500, and then either leave the parameter at 0.5 or change it to either 0.25 or 0.0. Because the vaccine-seeking Population 1 is more likely to maintain NPIs than Population 2, Population 2 is sometimes given a lower lockdown parameter. Vaccination continues in Population 1 after day 500. NPIs were surprisingly important in determining long-term rates of disease.

**Table 3 vaccines-12-00532-t003:** The effect of reducing NPIs starting on day 501 on vaccine-era case rates. Average duration of immunity is 135 days, and R_0_ = 2.87. The population consisted of 68% Population 1 (seeks vaccination) and 32% Population 2 (refuses vaccination). Each column heading shows the lockdown parameter for Population 1 first and Population 2 second.

Vac/Day	Lockdown after 500 Days				
	0.5/0.5	0.5/0.25	0.25/0.25	0.25/0.0	0.0/0.0
0%	773.6	991.4	1353.4	1444.1	1625.9
0.25%	411.5	688.1	1105.6	1225.8	1436.9
0.50%	145.9	462.5	875.4	1029.5	1272.0
0.74%	42.5	312.8	684.2	863.7	1126.3
1.00%	19.0	198.0	518.8	714.2	978.6

### 3.7. Percent of Population Accepting Vaccination and Weakening of NPIs

Table 4 shows that abandoning or weakening NPIs causes a surge in rates of disease regardless of whether Population 1 is high or low.

**Table 4 vaccines-12-00532-t004:** The interaction between abandoning NPIs and the percent of the population that seeks vaccination. Results are shown for the combination of Populations 1 and 2. R_0_ = 2.87, vaccination rate of Population 1 is 0.74%/day; average duration of immunity is 135 days. At 501 days into the simulation, both Populations 1 and 2 either leave their lockdown parameter at 0.5, reduce it to 0.25, or reduce it to 0.0.

Pop. 1/2	Lockdown after 500 Days		
	0.5	0.25	0.0
100%/0%	5.6	195.6	760.9
68%/32%	42.5	684.2	1126.3
50%/50%	198.1	898.5	1277.9
25%/75%	501.2	1148.9	1458.0
0%/100%	773.6	1353.4	1625.9

### 3.8. Weakening of Interest in Getting Booster Vaccinations

Just as usage of NPIs may weaken over time, the desire to get booster vaccinations may also fade. Table 5 simulates a Population 1 that maintained a high vaccination rate for the first year after vaccine introduction, but then experienced an exponential decline in vaccination rates after that.

**Table 5 vaccines-12-00532-t005:** Effect of loss of interest in booster vaccinations. R_0_ = 2.87, Population 1 is 68%, initial vaccination rate of Population 1 is 0.74%/day; average duration of immunity is 135 days, lockdown parameter maintained at 0.5. After the first year of vaccination, the vaccination rate of Population 1 declines by 0%, 0.1%, 0.25%, 0.5%, or 1.0% per day for the rest of the simulation. The last four rates of decline reduce vaccinations by half in 693, 277, 138, and 69 days, respectively. The table cells show cases/thousand/year in years 2–5.

	Vaccination Decline/Day				
	0.0%	0.1%	0.25%	0.5%	1.0%
Case Rate	42.5	256.4	483.8	608.6	673.9

### 3.9. Evolution of a More Transmissible Variant

Table 1, Table 2, Table 3, Table 4 and Table 5 assumed that the pathogen retained its “2020” R_0_ of 2.87. However, later dominant variants have been more transmissible. R_0_ for the delta variant has been estimated at 5.08 [17], and the omicron variant’s R_0_ has been estimated at 9.5 [18]. More transmissible variants were simulated by increasing R_0_ from its original value of 2.87 to a higher target R_0_ 500–600 days into the simulation and then keeping R_0_ at this higher level for the rest of the simulation. Even small increases in R_0_ had a pronounced effect on the post-vaccine case rates (Table 6), and a vaccination rate of 2.5% of the susceptibles per day had to be added as well.

**Table 6 vaccines-12-00532-t006:** Ability of several vaccination rates (percent of Population 1 susceptibles vaccinated/day) to control case rates as the base R_0_ of 2.87 was increased to various final R_0_s between days 500 and 600 of the simulation. Average duration of immunity is 135 days, 68% Population 1 and 32% Population 2, lockdown parameter maintained at 0.5 after 500 days.

Vac/Day	R_0_ after 600 Days			
	2.87	3.5	5.0	10.0
0%	773.6	1081.5	1520.7	2016.7
0.25%	411.5	770.8	1302.9	1902.0
0.74%	42.5	334.1	922.2	1689.7
1.00%	19.0	175.0	762.8	1590.3
2.50%	3.2	11.6	77.3	1155.4

### 3.10. Changes in the Average Duration of Immunity

All previous tables used average durations of immunity that were constant at 135 days. Although increasing the pathogen R_0_ usually caused a drastic increase in case rates, longer durations of immunity reduced the damage done by more transmissible variants:

**Table 7 vaccines-12-00532-t007:** Effects of different average durations of immunity (days) on case rates in a population consisting of 68% Population 1 and 32% Population 2, with a daily vaccination rate of 0.74% of the susceptibles/day. Lockdown parameter was maintained at 0.5 after 500 days.

Immunity Duration (Days)	R_0_ after 600 Days			
	2.87	3.50	5.00	10.00
50	893.0	1802.9	3068.5	4566.2
135	42.5	334.1	922.2	1689.7
250	9.2	45.6	332.8	832.3
500	2.0	7.0	81.2	302.8
750	0.5	2.6	27.5	154.8

### 3.11. Disease Exposure Reduces the Infectivity of the Virus

The Discussion will point out that the unmodified SEIIS model predicts far higher case rates than appear to be observed in the US in early 2024. One hypothesis to resolve this discrepancy is that repeated infections somehow alter the immune system to make infections more difficult. The infection experiences of Population 1 and Population 2 (with and without vaccination) were different and greatly affected by the R_0_ of the pathogen:

**Table 8 vaccines-12-00532-t008:** Percent of the population experiencing cases or vaccination when there is no effect of cases or vaccinations on R_0_. Pop. 1 = 68%, Pop. 2 = 32%, vaccination rate = 0.74% of Pop. 1 susceptibles/day, NPIs of 0.5 maintained. Column headings show the R_0_ reached 600 days into the simulation.

Population	R_0_ = 2.87	R_0_ = 3.5	R_0_ = 5.0	R_0_ = 10.0
Pop. 1	776.3%	825.0%	946.0%	1124.4%
Pop. 2	118.6%	298.8%	652.9%	1014.3%

High-R_0_ infections produce a greater exposure to virus antigens. This explains the results in Table 9. Recall that “Exposure_50_” is the number of cases or vaccinations that reduces the R_0_ by 50%, so the lower the value of Exposure_50_ is, the more rapidly cases and vaccinations reduce the ability of the virus to infect the population.

**Table 9 vaccines-12-00532-t009:** Final case rates when R_0_ is reduced by the percent of the population experiencing cases or vaccination. Exposure_50_ = percent exposure of the population that produces a 50% reduction in R_0_. Pop. 1 = 68%, Pop. 2 = 32%, vaccination rate = 0.74% of Pop. 1 susceptibles/day, NPIs of 0.5 maintained, duration of immunity is 135 days. Column headings show the R_0_ reached 600 days into the simulation.

Exposure_50_	R_0_ = 2.87	R_0_ = 3.5	R_0_ = 5.0	R_0_ = 10.0
No Effect	42.5	334.1	922.2	1689.7
1000%	7.3	36.0	333.4	1196.8
500%	3.0	10.9	131.3	845.3
250%	0.8	3.2	44.2	400.9
100%	<0.1	0.3	4.5	58.0
50%	<0.1	<0.1	1.0	19.4

### 3.12. Disease Exposure Increases the Average Duration of Immunity

Table 7 varied the duration of immunity but kept it constant within each simulation. Table 10 began all simulations with a mean duration of immunity of 135 days, but then allowed the duration of immunity to be increased by repeated exposure to virus antigens by both cases and vaccinations. Every 1% of the population that either contracted the disease or was vaccinated increased the mean duration of immunity by 0, 0.01, or 0.05 days. These calculations were separate for Population 1 and Population 2, so generally Population 1 ended with a longer mean duration of immunity because only Population 1 received vaccination. The rationale for this series of simulations will be explained in the Discussion.

**Table 10 vaccines-12-00532-t010:** Effect on case rates in years 2, 3, 4, and 5 of larger R_0_ values and an increasing average duration of immunity. Population consisted of 68% Population 1 and 32% Population 2, with a vaccination rate of 0.74%/day. Lockdown parameter was maintained at 0.5 after 500 days. Immune duration was 135 days. Increases in immune duration were 0, 0.01, or 0.05 days per cumulative percent of the population that had experienced cases or received vaccination.

Effect of Cases or Vaccination on Immune Duration	R_0_ after 600 Days			
	2.87	3.50	5.00	10.00
No Effect	42.5	334.1	922.2	1689.7
+0.01 days per %	2.7	9.6	99.4	299.1
+0.05 days per %	0.1	1.3	14.0	63.7

Table 11 examines the R_0_ = 10.00 column in Table 10.

**Table 11 vaccines-12-00532-t011:** Cumulative cases and vaccinations (over the entire 5 years of the simulation) from the “R_0_ = 10.00” column of Table 10, plus the average percentage of the population (both Population 1 and Population 2) that is made up by those with disease-induced immunity (Diim) and vaccine-induced immunity (Viim).

Effect of Cases	Cases	Vacs	Av. Diim %	Av. Viim %
No Effect	8.77 × 10^6^	2.07 × 10^6^	50.2%	12.0%
+0.01 days per %	2.41 × 10^6^	1.61 × 10^6^	40.3%	39.5%
+0.05 days per %	1.15 × 10^6^	1.03 × 10^6^	35.7%	33.1%

### 3.13. R_0_ Reduction or Increasing Duration of Immunity Can Counter Negative Developments

A final way to illustrate the powerful effect of R_0_ reduction or increasing the duration of immunity is to start with a base set of parameters describing a relatively controllable epidemic, progressively add negative developments such as dropping NPIs, and estimate how well R_0_ reduction and increasing duration of immunity could counter them. The base situation was that R_0_ remains at 2.87, the lockdown parameter remains at 0.5, vaccination rate of Population 1 remains at 0.74% of the susceptibles/day, and the duration of immunity is constant at 135 days. 

In Table 12, the following alterations to this base condition are simulated:A.R_0_ rises from 2.87 to 10.0 between 500 and 600 days.B.All NPIs are abandoned after 500 days.C.The vaccination rate of Population 1 declines at 1%/day starting one year after vaccination begins. The vaccination rate half-life is 69 days, and the decay of interest continues to the end of the simulation.D.R_0_ is reduced by infection with an Exposure_50_ of 50%.E.Immune duration increases by 0.05 days per percent of the population experiencing cases or vaccination.

The reduction in case rates in the last two lines shows the effect of allowing previous infections and vaccinations to reduce R_0_ and increase the duration of immunity.

**Table 12 vaccines-12-00532-t012:** Effect of developments A–E on case rates/thousand/year in years 2, 3, 4, and 5.

Condition	Case Rate
Base	42.5
Base + A	1689.7
Base + A + B	2098.7
Base + A + B + C	2252.5
Base + A + B + C + D	496.4
Base + A + B + C + E	221.7

## 4. Discussion

These simulations disclosed that even if a vaccine produces immediate sterilizing immunity against all COVID-19 variants, many common situations can impair its effectiveness. Success was measured by COVID-19 case rates in the period from 1 to 5 years following vaccine introduction. Independent variables considered were the percentage of the population that accepts the vaccine, the speed of vaccine distribution, maintenance of NPIs such as masking even after vaccines are in use, declining interest in getting booster vaccines, the development of more transmissible variants, and the duration of immunity. 

Case rates are expressed in terms of COVID-19 cases/thousand/year in the period from 1 to 5 years after the introduction of the vaccine (the red box in Figure 1). These case rates include the whole population (vaccine-accepting Population 1 + vaccine-refusing Population 2). A case rate of 9.2 (for example) indicates that the average person in this combined population has a 0.92% chance of a COVID-19 infection every year; a 2252 case rate means the average individual will suffer from 2.3 COVID-19 infections/year.

Many simulations predicted very high case rates, but these high rates were never due to a failure of the hypothetical vaccine itself nor an evasion of the immune response by a new variant. The simulated “ideal” vaccine always induced sterilizing immunity against every variant. The problems lay in factors like the speed of vaccine distribution, the discontinuation of NPIs, the speed of the spread of the disease in unvaccinated individuals, etc.

Many of the simulation results were expected, but there were also surprises. This Discussion will focus on the unexpected outcomes.

The speed of vaccine distribution is used many times in this report. As stated in the Methods, US CDC data on cumulative full vaccination [4] implies that during the US vaccination drive in the first half of 2021, Population 1 in the US was 68% of the total, and Population 2 was 32%. Also, there appeared to be a vaccination completion rate of 0.74%/day of the Population 1 susceptibles (which would be about 0.5%/day of the susceptibles in the whole population).

Table 1 shows that if the whole population (100% column) can be vaccinated at 0.74% of the susceptibles per day or faster, an epidemic with an R_0_ of 2.87 and vaccine-induced immunity of 135 days or longer can be essentially ended. The same is true if 75% of the population accepts vaccination, but then the vaccination speed must be 1.0% per day or greater. Greater unvaccinated portions of the population preclude ending the epidemic. It is noteworthy that with 68% of the population accepting vaccination and a vaccination rate of 0.74%/day (the default conditions), the case rate will still exceed the “Fauci limit” of 10 cases/thousand/year, even with a low R_0_ of 2.87. 

Table 1 shows the results for the combination of Population 1 (accepts the vaccine) and Population 2 (refuses the vaccine). It might be supposed that at least Population 1 should be a well-protected minority. It could be, but only if it can escape exposure to Population 2. For the table cell at the lower right, with Population 1 at 25% and a vaccination rate of 1% per day, the case rate (cases/thousand/year) for Population 1 is 252.1 and for Population 2 is 518.9. Population 1, despite its vaccination, is inundated with infections coming from the larger Population 2. The first column shows that if there was no Population 2, Population 1’s case rate with a 1% vaccination rate would be only 2.1 cases/thousand/year. These patterns are all expected, but the extent to which Population 1 is contaminated by cases from Population 2 on the right side of Table 1 might be a surprise. For a 0.74%/day vaccination rate, if Population 1 declines from 75% to 25% of the population, the overall case rate increases by a factor of 21 due to cases coming from Population 2. Vaccine hesitancy is a problem for the whole population, not just for the vaccine refusers.

While mixing randomly with Population 2 hurts Population 1, the mixing benefits Population 2. Table 1 shows that with no vaccination at all (first line of the table), both Population 1 and Population 2 have a high case rate of 773.6 cases/thousand/year. However, if the vaccination rate is 0.0074% of Population 1 susceptibles/day, Population 1’s final case rate is 33.1 and Population 2’s is 62.6 cases/thousand/year. In other words, Population 2, without ever getting vaccinated itself, has cut its case rate by a factor of 12.4 by being associated with a population that is being vaccinated and therefore has a much lower case rate. The intelligent vaccine refuser can be right to hope that everyone *else* gets the vaccine.

Table 2 shows the synergistic effects of the vaccination rate and the duration of immunity. If immunity lasts only a short time, there will not be enough time to vaccinate a large portion of the population before the immunity begins to expire. The right side of Table 2 shows that immunity with an average duration of 500–1000 days can nearly end the epidemic even if the vaccination rate is moderate. Unfortunately, vaccinal immunity appears to have such a short duration [21] that even high vaccination rates still allow a substantial case rate (left side of Table 2). The duration of immunity is a vital consideration and will be revisited when Table 7, Table 10, Table 11 and Table 12 are discussed.

NPIs like masking and social distancing, far from being replaced by vaccination, are essential to allow vaccination to have its best effects. Table 3 assumes that Populations 1 and 2 maintain a lockdown parameter of 0.5 until day 500, about 4.5 months after the vaccine is introduced, then they may reduce their adherence to NPIs. For a vaccination rate of 0.74%/day (for Population 1), if both Populations 1 and 2 maintain a lockdown parameter of 0.5 after day 500, the case rate is a low 42.5 cases/thousand/year. If Population 2 drops its lockdown parameter to 0.25, the case rate multiplies by more than seven times. If Population 1 also drops to 0.25, the case rate multiplies by 16 times over the 0.5/0.5 rate. If both populations drop the lockdown parameter to zero, the case rate is 26.5 times the 0.5/0.5 value, even if vaccination continues. This is a striking result. Even with a low R_0_ of 2.87 and rapid vaccination with an all-variant vaccine, even a partial dropping of NPIs results in a surge in cases.

It might be supposed that a high vaccination rate could shield the population from the dropping of NPIs because vaccination could replace NPIs. The simulation results do not support this prediction. Table 3 shows that vaccination plus NPIs produces much lower case rates than NPIs alone or vaccination alone can produce. In Table 3’s upper left-hand corner, NPIs without the vaccine produce a case rate of 773.6; in the lower right-hand corner, a rapid vaccination rate without NPIs produces a rate of 978.6; with both vaccine and NPIs working together (lower left of Table 3), the case rate can be as low as 19.0 cases/thousand/year. 

The reason for the importance of NPIs is straightforward. In the SEIIS model, the number of new cases per iteration is proportional to the number of susceptibles (Equation (1)). If a population has a lockdown parameter of 0.5, it is protecting half of its susceptibles from infection. If it drops its lockdown parameter to zero, it doubles the number of susceptibles the virus can infect, which will quickly double the number of new infections. These results recall the conclusion of several simulation papers in the Introduction that emphasized the importance of NPIs to the success of the vaccine. 

However, Table 3 used a low R_0_ of 2.87, so the R_0_ could almost be brought below 1.0 by a lockdown parameter of 0.5. If Table 3 is given an R_0_ of 5.0, the zone of relative success at the lower left of Table 3 disappears and all the case rates become higher. With an R_0_ of 5.0, the equivalent case rate for 1% vaccination and a lockdown parameter of 0.5/0.5 is 762.8 cases/thousand/year; the case rate for no vaccination and no NPIs is 2016.1 cases/thousand/year. A higher R_0_ reduces the effect of both NPIs and vaccination.

Table 4 also shows the powerful effect of abandoning NPIs. Where Population 2 is zero (first row), cessation of NPIs causes case rates to multiply by more than 100 times. This happened because Population 1 was benefitting from both vaccination and NPIs, and its case rate was extremely low, allowing a big increase when NPIs are dropped. However, where Population 1 is 50% or lower, cases that Population 1 caught from Population 2 raise case rates in the lower part of the table. In the last row of the table, dropping NPIs only increases cases by a factor of 2.1. This is not good news. It occurs not because the “0.0” column had low case rates, but because the “0.5” column had high case rates. When case rates are already high, they cannot increase by a big multiple.

As of 10 May 2023, 81.4% of the US population had received one vaccine dose, 69.5% had received two doses, and only 17% had received at least one booster dose of the bivalent vaccine [4]. Table 5 explores the consequences of this fading interest in keeping vaccinations up to date. When Population 1 keeps getting vaccinated, its case rate is 33.1 cases/thousand/year, and Population 2’s rate is 62.6. When Population 1 undergoes a 1% decline per day in its vaccination rate (vaccination half-life of 69 days), the final case rates for Populations 1 and 2 are 666.0 and 690.7 cases/thousand/year, both high and almost equal. Population 2’s case rate has gone up by a factor of 11 due to the vaccine negligence of Population 1, not due to any change in Population 2’s behavior. This provides another example of the linkage between Populations 1 and 2.

Table 6 shows the effect of variants with increased R_0_ values on case rates. Even an increase of R_0_ from 2.87 to 3.5 increases the post-vaccine case rate by a factor of almost 8 with a fast vaccination rate of 0.74%/day. If the increase is from an R_0_ of 2.87 to 5.0, the case rate goes up 21.7 times for the 0.74%/day vaccination rate. Also, the effectiveness of increasing the rate of vaccination declines as R_0_ rises. A very high vaccination rate of 2.5% of the susceptibles/day can drop the case rate by 90% (over the 1% vaccination rate) when the R_0_ = 5.0, but when the R_0_ reaches 10, even this extreme vaccination rate only drops the case rate by 27% below the level produced by a 1% vaccination rate, and only 43% below a *zero* vaccination rate. High R_0_s multiply case rates, and realistic vaccination rates (even with the simulated universal coronavirus vaccine) cannot keep up. 

Another startling conclusion from Table 6 is that once R_0_ reaches a level between 5.0 and 10.0, it seems that vaccination has only a weak effect on case rates. When R_0_ = 2.87, a vaccination rate of 0.74%/day lowers the case rate by 95% compared with the case with no vaccination. Where R_0_ = 5.0, vaccination at the same rate only produces a reduction of 40%. When R_0_ = 10.0, a 0.74%/day vaccination rate only lowers the case rate by about 17% over the case with no vaccination at all. To cut the R_0_ = 10.0 case rate in half would require an impossible sustained vaccination rate of 3.3% of Population 1 susceptibles per day. A realistic range of R_0_s almost neutralizes the ability of even the hypothetical pan-coronavirus vaccine to reduce cases.

Why can variants with high R_0_s escape control by the vaccine? The escape does not occur because high-R_0_ variants evade the immune response; the simulated vaccine remains 100% effective against all variants. The higher case rates result from the fact that the disease spreads faster than vaccinations can be given. In a Population 1 simulation with R_0_ = 2.87 and a vaccination rate of 0.74%/day, at the end of year 5, Population 1 had experienced 0.69 × 10^6^ cases and 4.6 × 10^6^ vaccinations. That is, with a 2.87 R_0_, the number of vaccinations dwarfs the number of new cases. On the other hand, when R_0_ was 10, those two cumulative numbers for Population 1 were 5.6 × 10^6^ cases and 2.1 × 10^6^ vaccinations. A higher R_0_ has drastically increased the number of new cases, which now exceeds the number of vaccinations. In an environment with high-transmission variants, one might imagine that the disease and the vaccine are in a race to convert susceptibles to either new cases (by contracting the disease) or to vaccine-induced immunes. The relative advantage of the vaccine in this race falls in Table 6 as the R_0_ increases.

Table 7 shows the importance of the duration of immunity when high R_0_ values are present. Longer durations of immunity are always helpful because they give even moderate rates of vaccination the chance to cover more of the population before vaccinal immunity expires. An immune duration of 135 days allows a variant with an R_0_ of 10 to give the average member of the population 1.7 COVID-19 cases per year (case rate of 1689.7 cases/thousand/year). With an immune duration of 750 days, the same high-transmission variant can only give the average member of the population a 15% chance of one COVID-19 case per year (case rate of 154.8 cases/thousand/year).

A major paradox of the results in Table 1, Table 2, Table 3, Table 4, Table 5, Table 6 and Table 7 is that they predict a much worse COVID-19 epidemic than actually exists in 2023–2024. In the US in early 2024, rates of compliance with NPIs and booster vaccinations were extremely low, and new variants such as JN-1 were present. The JN-1 variant first appeared in the CDC “Nowcast” report on 28 October 2023, and by 24 February 2024 had accounted for 96.4% of specimens sequenced [23]. The 2024 COVID-19 environment in the US certainly seems to be active and dynamic. 

However, while US COVID-19 hospitalizations rose and then fell in early 2024, the data do not resemble anything like the omicron surge of early 2022. In that surge, there was a peak of 150,000 hospitalizations/week in January 2022. In contrast, there were only 15,141 hospitalizations/week by 2 March 2024 [24]. Using the ratio of 14.46 cases/hospitalization mentioned in the Methods, the estimate of the case rate on 2 March 2024 is 218,939 cases/week = 34.5 cases/thousand/year for the US population of 330,000,000. This moderate case rate seems more in line with the present American experience than the predicted rates of over 1000 cases/thousand/year that are common in Table 3, Table 4, Table 5, Table 6 and Table 7. So, why are COVID-19 case rates relatively low in the US despite the cessation of NPIs, low vaccination rates, and fast-spreading new variants? While a definitive answer is unclear, there are some possible explanations.

One explanation for moderate case rates is that the R_0_s of the new variants are lower. This would produce lower rates of infection, but it is difficult to explain how viral evolution would favor a change to a lower R_0_. 

On the other hand, perhaps strengthening host secondary immunity has made viral R_0_s lower, perhaps by lowering the probability that an encounter with the virus will cause an infection. This hypothesis was explored in Table 8 and Table 9.

A third explanation in the US popular press is that COVID-19 symptoms have become significantly milder [25], and so even though the rate of new cases may be high, most cases are unreported or ignored. This hypothesis is not supported by the data on US hospital admissions. In the 20 weeks from 14 May 2022 through 24 September 2022, in the aftermath of the omicron surge in January 2022, average new US hospital admissions due to COVID-19 averaged 33,970/week. In the 20 weeks from 30 September 2023 through to 10 February 2024, the average was 22,317 admissions/week [24]. This rate of admissions is lower than in 2022, but the disease in the US in late 2023 and early 2024 was still severe enough to cause tens of thousands of Americans to visit an emergency room each week. Also, the average number of deaths/week from COVID-19 in January and February 2024 was 1791.9. This is about 70% of the US weekly death toll in the second half of 2022. The 2024 death rate is lower, but almost 2000 deaths per week is not trivial.

Table 8 and Table 9 explore the possibility that high rates of past disease have strengthened the immune systems of potential patients so that the virus cannot infect them as readily. In practical terms, the R_0_ of the virus has been reduced. For example, for a variant with an R_0_ of 10, the simulated case rates in years 2–5 fell from 1689.7 cases/thousand/year with no R_0_ reduction to fewer than 100 cases/thousand/year with a strong reduction. As simulated, this hypothesis does not include any shortening of the average immune duration of 135 days.

Another possibility, which would be possible to demonstrate experimentally, is that infections and vaccinations have left R_0_ unaltered, but have lengthened the average duration of COVID-19 immunity. Recall that the data from Menegale et al. [21] implied that the average duration of immunity offered by the first COVID-19 vaccine dose was less than 135 days, but the average duration offered by the second dose was less than 224 days, significantly longer. This suggests that the duration of vaccinal immunity might be increasing, perhaps due to exposure to viral antigens.

Table 10 explores the consequences of increasing immune duration. Using model results on the number of infections and vaccinations experienced by Population 1 and Population 2, the model increases the duration of immunity as these numbers build. In the “R_0_ = 10.00” column of Table 10, the final duration of immunity for the “+0.01” line had risen from 135 days to 708 days for Population 1 and 610 days for Population 2. These lengthened periods of immunity drastically reduced the case rates for the combined (1 + 2) population from 1689.7 cases per thousand/year (without lengthening) to 299.1 (with lengthening), even for a variant with an R_0_ of 10. For a lengthening of 0.05 days per percent, an R_0_ of 10 only produced a rate of 63.7 cases/thousand/year. Case rates for lower R_0_s sometimes even met the “Fauci standard” of being less than 10 cases/thousand/year. 

In order to determine why these sharp case reductions occurred, Table 11 examines some details of the R_0_ = 10 column in Table 10. First, as the duration of immunity increases, cases decrease sharply, and immunity shifts from dominance by disease-induced immunity to a greater role for vaccine-induced immunity. In other words, increasing duration of immunity promotes immunity without disease.

Table 12 explores the hypothesis of lengthening the duration of immunity in another way. It simulates a relatively controllable “base” epidemic and then progressively adds more and more negative developments such as arrival of high-transmission variants, dropping NPIs, and a Population 1 that loses interest in getting vaccine boosters. Both R_0_ reduction and immune duration enhancement were able to cut cases sharply, and an increase in immune duration reduced cases to about 10% of the rate before the enhancement was added.

While the simulations show that R_0_ reduction or immune duration enhancement could reduce case rates, determining whether one or both of these theories are correct would require more direct evidence of disease effects on R_0_ and immune duration. 

To conclude, this study verified many unsurprising results (e.g., faster vaccination rates and less vaccine hesitancy can reduce long-term case rates). However, some findings were surprising. 

First, the success of even a universal coronavirus vaccine that confers immediate sterilizing immunity can be limited by many common conditions. Just to take two examples, Table 1 shows that a decrease in Population 1 from 68% to 50% (with a 0.74%/day vaccination rate) can increase the long-term case rate by almost five times. If the vaccination rate also drops from 0.74%/day to 0.5%/day, the increase will be almost eight times. 

Another striking example is the effect of the reduction of NPIs in Table 3. It was remarkable that even if vaccination continues at 0.74%/day, decreasing the lockdown parameter (in both Population 1 and 2) from 0.5 to 0.25 results in a 16-fold increase in case rates, and abandoning NPIs entirely increases case rates by 26.5 times. The R_0_ in both Table 1 and Table 3 was a constant 2.87. While this example was not in the tables, if we combine a dropping of NPIs to zero with an increase of R_0_ from 2.87 to 5.00 after day 500, the long-term case rate multiplies by almost 40 times.

The second major finding was the paradox that despite the end of NPIs, declining interest in getting booster vaccinations, and the presence of new, fast-spreading variants, COVID-19 case rates were relatively low in the US in early 2024. A common theory quoted in the American popular press is that immunity to COVID-19 is high because large numbers of Americans have been vaccinated or have had the disease [25]. However, Table 6 and Table 7 show that high-R_0_ variants could not be constrained by an immunity with an average duration of only 135 days. If we assume the exponential model of immune waning and a mean duration of vaccinal immunity of 135 days, the majority of the US population that got vaccinated for the last time in early 2022 is now 5.4 average immunity durations past their last vaccination (about 7.8 half-lives). Less than 1% of their vaccinal immunity should be left.

To answer the question of why COVID-19 case rates are so low in the US in early 2024, this paper hypothesizes that past exposure to the virus is either making it more difficult for the virus to infect or increasing the duration of immunity (Table 9 and Table 10). Of course, as already observed, verifying that this is true would demand direct observation of resistance to infection and changes in the duration of immunity, and elucidation of the mechanisms through which these effects occur. 

### Limitations of the Study

There are reasons to accept these results with caution. First, the SEIIS model used here is a simple compartmental model. It did not simulate the age or health structure of the US population, and it had no mortality. There was no seasonal increase in infections. It assumed that every organism in the susceptible compartment reacts the same way, and the same was true for every organism in the immune compartment. These are important simplifications. One can imagine that susceptibles that had been infected by the virus before might react differently to new contact with the virus than susceptibles that had never been infected. An individual entering immunity for the first time might have a different immune waning rate compared with an immune individual who has had multiple past infections. One reason these simplifications might have less of an impact is that the model output was the average case rate over the final four years of the simulation. Therefore, it would be adequate to estimate a general level of disease, but perhaps not to predict each individual outbreak. 

Also, we must consider another general weakness of simulations. Assuming that they include the proper relationships and parameters, simulations estimate what would happen if a set of conditions is continued for the length of the simulation. If real-world case rates were escalating drastically, the parameters of a real system would tend to react and change—NPIs would hastily be restored, vaccination speeds would go up, and some Population 2 individuals would shift to Population 1. The proposed increasing duration of immunity is an example of a change that would moderate results.

The model also implicitly assumed that members of the whole population were interacting randomly with one another. The complex history of COVID-19 in the US has many examples where cases would flare up in a region, but then decrease without spreading nationwide. For example, at the height of the pandemic, the US popular press mentioned a “two-month COVID cycle”, in which cases tend to increase for two months and then recede, for unknown reasons [26]. 

The SEIIS model used here is simple, but even a simple model can lead to insights. If the data do not match the model, we know we have to look for other explanations than the ones the model supplies.

## 5. Conclusions

The first major conclusion of the paper is that many factors aside from vaccine efficacy can affect the success of a COVID-19 vaccination campaign. Despite the fact that the simulated vaccine always had 100% efficacy against infection, it frequently allowed high rates of disease, particularly when non-pharmaceutical interventions such as masking were dropped, high-transmission variants were present, and vaccine hesitancy was moderate or severe. The second major conclusion was that the basic model could not explain the seemingly low rates of disease in the US in 2023–2024. The paper hypothesized that some factor (probably connected with long-term adaptation of the human immune system to contact with viral antigens) is reducing the COVID-19 case rates.

## Data Availability

All data used in this study were gleaned from published sources that are cited in the References. The simulations were generated using an Excel spreadsheet. The author will provide a copy of the spreadsheet to interested parties upon request.

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
