# Peer review of "The Limitations of a Hypothetical All-Variant COVID-19 Vaccine: A Simulation Study"

_vaccines, 2024, doi:10.3390/vaccines12050532_

Round 1

Reviewer 1 Report

Comments and Suggestions for Authors

The author produces an interesting article, in which he tries to rationalize the experiences gathered during the recent pandemic and the incessant debate that resulted from it using a theoretical model to evaluate the epidemiological course of an ideally perfect vaccine.

1.     The theoretical model introduces some factors. They all seem relevant. The authors use a compartmental model. On line 130 they indicate the acronym, without explaining the meaning (which they do a few lines below). I would suggest anticipating the meaning and briefly explaining what a compartmental model is.

2.     The author explains in detail the choices made in the various steps of the model. The choices are based on literature data.

3.     The results investigate various possibilities and demonstrate that even an ideal vaccine must deal with situations that limit its effectiveness. The Discussion is mainly focused on unexpected results; this choice is correct. Personally, I have the impression that the author chose a simple compartmental model precisely to obtain unexpected results. However, he correctly points out this fact in the limitations of the study; the chosen model is too simple to effectively simulate reality, but it has obvious teaching effectiveness and indicates why even an ideal vaccine does not have magical power.

Author Response

I thank the reviewer for his/her comments. Changes I have made to the text are highlighted in red in the revised copy of the MS.

In the revised MS, I have more fully explained what a compartmental model is as I introduce the compartments of the model. These changes are highlighted in red in the post-review version of the MS.

The model was simple, but I did not choose it to obtain results that differed from the the real world. When the model predicted much higher case rates than are actually being experienced, I had to ask why. After all, the model makes its predictions based on a simple continuation of infection rates, duration of immunity, etc., that existed at the beginning of the pandemic, when all the calculations of infectivity and immune duration were done. If the actual rates of disease in 2024 are much lower than those numbers would predict, some kind of change in the operation of the disease must have taken place. I proposed two possibilities--increasing duration of immunity, and decreasing infectivity of the virus. I don't know that these changes actually occurred, but they do bring the rates of disease more into the observed range.

Reviewer 2 Report

Comments and Suggestions for Authors

This is an interesting paper that can be used to understand policies in the are of covid19 prevention and vaccination. Please answer the following questions.

ABSTRACT

1.      In the abstract explain the SEIIS acronym, do the same in the text the first time it is introduced.

MATERIAL AND METHODS

2.      Explain briefly in the material and methods what a SEII is so that unfamiliar readers can understand. (There is some comment in the discussion section).

3.      Did the you  did anay sensitivity analysis to see how changes in the assumptions influced the outcome of the models?

DISCUSSION

4.      In the discussion comment on whether you think your findings are limited to COVID-19 only or could be extrapolated to other vaccines such as influenza.

5.      The model presented doESn’t consider factors such as seasonal changes in mortality rates and virus tramsmission. Please discuss how this can affect the model.

6.      Discuss how different lvels of compliance with NIPs can impact the outcome.

7.      In the discussion could you compare SEISS models with other epidemiological models, and explain in your opinion the advantages and disadvantages.

8.      Do you think that this model and its conclusions can be applied to all the world or is more limited to de United States?

SUPPLEMENTARY MATERIALS

9.      Include the Excels you used as supplementary material so interested readers can understand the data.

Author Response

I thank the reviewer for his/her comments. Changes I have made to the text are highlighted in red in the revised copy of the MS.

1. Abstract: I have included an explanation of the compartments of the model in the abstract

Materials and Methods:

2. I have explained the compartments more fully in the Materials and Methods.

3. I did not do a formal sensitivity analysis, but in a sense the whole paper is an exploration of parameter space. For example, finding that a 50% reduction in NPIs could cause a 16x increase in cases for the default conditions (Table 3) was a surprise, as was the enormous increase in case rates caused by seemingly small changes in R0 (e.g., a 22-fold increase in case rates by increasing the R0 from 2.87 to 5.0 in Table 6). Sensitivity of the system to relatively small changes in parameters are noted many times in the MS.

Discussion:

4. Yes, I think the model (with necessary changes in parameters) could apply to any communicable disease. Compartmental models such as this have been used in epidemiology since the work of Kermack and McKendrick in the 1920s, and I have included a citation of that paper in order to convey that use of simulation models has a long tradition in epidemiology.

5. Can the omission of seasonal changes in transmission affect the model outcome? This would be a big problem if I were attempting to predict the detailed course of the epidemic, but it is less of a problem with this study because the result is the average case rate during the four-year period at the end of the study period. For this prediction, estimating a general level of transmission is adequate. If I were trying to predict each peak in real time so that public health authorities were ready with large supplies of the vaccine, my approach would probably be inadequate. I have placed these ideas into the Discussion.

6. Yes, NPIs are surprisingly important for determining the long-term case rate, and I make this clear with extensive discussion in Section 3.6 and Tables 3 and 4. The public may believe that arrival of the vaccine relieves them from worry about NPIs such as masking and social-distancing, but this is not the case if infectivity of the virus remains high, vaccinal immunity is short-lived, and efficacy of the vaccine is low.

7. The different types of compartmental models are named after their compartments. The earliest was the SIR model = susceptible—infected—removed. Removed means that the victim either dies or becomes permanently immune, obviously not relevant for COVID-19, where vaccinal immunity is fleeting and death is infrequent. Another is SIS = susceptible—infected—susceptible, which describes COVID-19 better. I used SEIIS because it is known that an infected individual needs about five days to be infective, and then an infective individual only remains that way for five days (on the average). Given these facts, SEIIS seems to be the best choice.

8. Yes, to the extent that COVID and its variants are the same in different countries, the model conclusions would be applicable to other environments.

Supplementary Materials:

9.  I could include the spreadsheet I used, but the problem is that the spreadsheet is not very user-friendly. For example, say you want to make the population drop its NPIs on day 500. You would have to go down to row 500 in the NPI column and change the NPI variable from 0.5 to some other value. This would then be copied into the rows below. And then you’d have to remember that you did that as you did other manipulations. This forgetting of the changes I’ve made was a problem  for me, so I established the policy that I would only save the ”base” version of the spreadsheet. When I was finished with one set of experiments, I would quit and re-open it to be sure that there were no forgotten changes from previous experiments still active

But yes, I will include the spreadsheet in the Supplementary Materials.

Reviewer 3 Report

Comments and Suggestions for Authors

Dear Author,

The article is certainly well done and very interesting. The presentation of materials, methods, and results is well discussed. While the limitations are effectively elucidated in the discussion, I would recommend titling a section specifically as 'Limitations of the Study' for better emphasis. Additionally, while the utilization of press journal articles in the references is adept, I would encourage seeking out scientific literature whenever possible to enhance credibility.

Thank you sincerely for your efforts.

Best regards

Author Response

I thank the reviewer for his/her comments. Changes I have made to the text are highlighted in red in the revised copy of the MS.

I have entitled a section of the Discussion “Limitations of the Study,” as you suggested.